# Role of Narrow Band Imaging Technology in the Diagnosis and Follow up of Laryngeal Lesions: Assessment of Diagnostic Accuracy and Reliability in a Large Patient Cohort

**DOI:** 10.3390/jcm10061224

**Published:** 2021-03-16

**Authors:** Jacopo Galli, Stefano Settimi, Dario Antonio Mele, Antonio Salvati, Enrico Schiavi, Claudio Parrilla, Gaetano Paludetti

**Affiliations:** 1Institute of Otorhinolaryngology, Fondazione Policlinico Universitario Agostino Gemelli IRCCS, 00168 Rome, Italy; jacopo.galli@iol.it (J.G.); claudioparrilla@yahoo.com (C.P.); gaetano.paludetti@unicatt.it (G.P.); 2Institute of Otorhinolaryngology, Università Cattolica del Sacro Cuore, 00168 Rome, Italy; settimi.stefano90@gmail.com; 3Airway Surgery Unit, Pediatric Surgery Department, Bambino Gesù Children’s Hospital, 00165 Rome, Italy; antonio.salvati@opbg.net; 4Faculty of Medicine and Surgery, Università Cattolica del Sacro Cuore, 00168 Rome, Italy; enricoschiavi@gmail.com

**Keywords:** narrow band imaging, larynx, head and neck cancer

## Abstract

Background: The aim of this study was to assess diagnostic accuracy and reliability of narrow band imaging (NBI) in the differential diagnosis of laryngeal premalignant lesion, early cancers and recurrences. Material and methods: We enrolled 231 patients who underwent endoscopic examination with white light endoscopy (WLE) + NBI and divided them into two groups, group A, without previous radiochemotherapy and group B, with previous radiochemotherapy. When indicated, we performed surgical biopsies to evaluate sensitivity, specificity, positive predictive value (PPV), negative predictive value (NPV), accuracy and likelihood of endoscopic examination comparing WLE alone and WLE + NBI. Results: A positive NBI lesion, compared with a negative NBI lesion, had a 29.68 (group A) and 13.96 (group B) times higher probability to be histologically positive (i.e., confirmed) compared with WLE alone improving the diagnostic accuracy. In group A, the NBI mode showed excellent sensitivity (95.0%), which was higher than WLE 2 mode (77.5%). However, the greatest differences were recorded regarding specificity (96.8% vs. 40.6%). In group B, both NBI alone and WLE + NBI mode showed a 94.1% specificity compared with WLE alone, which had a maximum specificity of 85.3%. The mode comparison between NBI and WLE in both groups showed a statistically significant difference, with *p*-values <0.0001. Conclusions: NBI represents a reliable technology in challenging situations, especially in the context of post-radiotherapy or post-surgical mucosal changes showing a high NPV. NBI could reduce the number of unnecessary biopsies related to increased microvascular anomaly revelation, which could help to identify early-stage lesions suitable for minimally invasive surgery and, consequently, decrease hospital admissions.

## 1. Introduction

The mucosal surface of the upper aerodigestive tract (UADT) is an onset site of several pathologies with different etiology, pathogenesis and clinical presentation. Moreover, the UADT epithelium of the nasal and oral mucosa to the hypoglottic larynx shows pathological peculiarities, thus representing a challenging tract to explore. Considering that benign and malignant lesions represent some of the most frequent pathologies examined by the otolaryngologist, in the last decades a lot of new techniques were developed to examine the UADT [1]. Nowadays, the tract can be easily explored with traditional endoscopic techniques, but several studies and improvements in the endoscopic sector took place in the last years which increase diagnostic precision with simple and reliable equipment [2,3]. In fact, currently available diagnostic tools, including traditional endoscopes equipped with white light, are insufficient to unequivocally assess observed pathologies of the oral and laryngeal mucosa, hampering differential diagnosis of early neoplastic lesions and the extent of tumor infiltration [4]. In the last years, one of the most innovative and useful technologies is narrow-band imaging (NBI). NBI is an endoscopic optical imaging technique to enhance mucosal surface texture contrast and visualization of mucosal and submucosal vasculature increasing the diagnostic sensitivity on surface lesions or neoangiogenic patterns, which cannot be detected by regular white light endoscopy (WLE) [1]. NBI is based on the different light-absorbing properties of blood and mucosa; at this purpose, NBI filters produce only blue (415 nm of wavelength) and green (540 nm of wavelength) light, which visualize with different depth the mucosa, coloring superficial and submucosal vessels brown. This technology allows the clinician to assess the vascularization of mucosal lesions, increasing the diagnostic accuracy of traditional high-definition video laryngoscopy. NBI already showed its usefulness in early diagnosis of neoplastic lesions of the UADT [2,3,4,5] and in 2011, Ni et al. [6] proposed a classification based on the modification of mucosal vascularization observed with NBI allowing the in vivo differentiation of non-malignant and malignant laryngeal lesions. Furthermore, NBI is widely used for inflammatory pathologies of the head and neck in all age groups [7,8]. However, a precise diagnosis of mucosal lesions remains challenging and recent studies aimed at assessing its diagnostic accuracy in laryngeal lesions [9,10]. Considering the wide application of NBI in the head and neck area and the difficulties encountered by the physician with the number of different pathologies affecting these structures, the aim of our study was to assess the accuracy of NBI based on the differential diagnosis of laryngeal premalignant lesion, early cancers and tumor recurrences after radiotherapy in a large case series and single institution. Particular attention was paid to the negative predictive value (NPV) to assess its value in reducing numbers of surgical biopsies, especially in patients with unfavorable clinical conditions.

## 2. Materials and Methods

### 2.1. Design of the Study and Population

This was a single-center retrospective observational study carried out, between January 2017 and May 2019, in an Otorhinolaryngology unit at the Fondazione Policlinico Universitario Agostino Gemelli IRCCS in Rome (Italy). All outpatients who underwent a high-magnifying transnasal endoscopy with NBI technology as part of their physical examination, and patients older than 18 years affected by a macroscopic lesion of laryngeal mucosa, with endoscopic signs of cancer transformation or precancerous characteristics, were included. Patients who received diagnosis of benign lesions (such as laryngeal polyps, cysts, Reinke’s edema or vocal nodules), without signs of hyperkeratosis or papillomatosis, were excluded. We divided patients into two groups: patients without previous radiochemotherapy treatment (group A) and patients with previous radiochemotherapy treatment for laryngeal cancer (group B).

### 2.2. Instruments and Endoscopic Evaluation

Our outpatient clinic is equipped with a video rhinolaryngoscope and built-in NBI technology (CV-170; Olympus Medical System, Tokyo, Japan). The endoscopic examination was performed by a single experienced otolaryngologist with the patient in supine position. During the exam, a button allowed switching from white light endoscopy (WLE) to NBI. Ecocain^®^, (Molteni Dental srl, Milan, Italy) Lidocaine, 10 g/mL; dose: 2 sprays was administered 15 min before the procedure. Transnasal endoscopic images were reproduced on a 1080 p monitor and all exams were recorded for remote replay and review by a second examiner. Lesions assessed in WLE were classified as benign (0), uncertain (1) or clearly malignant (2), depending on the opinion of each examiner. Using NBI, laryngeal lesions were categorized according to the classifications proposed by Ni et al. [6] as follows: (0) non-malignant (pattern type I and II); (1) suspected leucoplakia and high-grade dysplasia or carcinoma (pattern type III, IV and V). In case of concordance between diagnoses of the two examiners, the lesion was classified as described. In case of disagreement, both examiners reviewed the video of the endoscopy again and a joint diagnosis was expressed.

### 2.3. Diagnostic Procedures

According to our clinical practice, if necessary and if patients’ clinical status allowed it, after endoscopic evaluation a biopsy of the lesion was planned; otherwise, patients were sent to close follow-up. Laryngeal biopsy in the operating room was foreseen by means of a surgical procedure under general anesthesia, using an operative microscope. Tissue samples were formalin-fixed and paraffin-embedded, then examined by a single expert pathologist. The histopathological evaluation was performed according to WHO classification as follows: benign lesion (0), low-grade dysplasia (1), high grade dysplasia (2), carcinoma in situ or invasive (3) [11]. Depending on histology, lesions classified as high-grade dysplasia (2) and carcinoma in situ or invasive (3) were considered positive, while lesions classified as benign lesions (0) and low-grade dysplasia (1) were considered negative.

### 2.4. Statistical Analysis

We evaluated sensitivity, specificity, positive predictive value (PPV), negative predictive value (NPV), accuracy, likelihood ratios and area under the ROC (Receiver Operating Characteristic) curves (AUC) in the following four modalities, according to possible diagnosis combinations:NBI: lesions classified in NBI, suspected as high-grade dysplasia or carcinoma (1), were considered as positive; lesions classified as non-malignant (0) as negative.WLE 1 + 2: lesions classified in WLE as uncertain (1) or clearly malignant (2) were considered positive; lesions classified as benign (0) as negative.WLE 2: only lesions classified in WLE as clearly malignant (2) were considered positive; lesions classified as benign (0) or as uncertain (1) were considered negative.WLE 1 + 2 + NBI: lesions classified in NBI, suspected as high-grade dysplasia or carcinoma (1), and at the same time classified in WLE as uncertain (1) or clearly malignant (2) were considered as positive; lesions classified in white light as benign (0) or classified in NBI as non-malignant (0) were considered as negative.

In addition, lesions for which no biopsy was performed, and which were stable or completely regressed after an endoscopic follow-up every 3 months for more than 18 months, were included in the statistical calculation as negative biopsies. An SPSS Statistics version 24 (IBM) database was used for statistical analysis. We used Chi-square test and ROC curves, and for the four modes listed above, AUC was also calculated, with the respective 95% confidence intervals.

## 3. Results

Between January 2017 and December 2019, 958 endoscopic examinations with WLE + NBI were performed at our institution, among which 196 patients met our inclusion criteria as they presented a macroscopic laryngeal lesion with suspicious endoscopic signs or potentially malignant transformation. One hundred and forty-five patients out of 196 (74.0%) were male and had a mean age of 60.5 ± 12.3 (range 26–93).

Group A included 156 out of 196 (79.6%) patients and 39 patients showed more than one lesion, for a total of 195 lesions. Group B included 40 out of 196 (20.4%) patients and 11 patients showed more than a single lesion, for a total of 51 lesions (Table 1).

### 3.1. Group A

Surgical biopsies were indicated and performed in 70 out of 195 (35.9%) lesions. Histology of Group A resulted as follows:
benign (0), *n* = 25 (35.7%) (Figure 1)low-grade dysplasia (1), *n* = 5 (7.1%)high-grade dysplasia (2), *n* = 12 (17.1%)carcinoma in situ or invasive (3), *n* = 28 (40.1%)

We also included six stable and one regressed lesion at follow-up as negative biopsies (mean follow-up of 23.33 ± 3.14 months). The values of sensitivity, specificity, PPV, NPV, accuracy and likelihood were then calculated and are shown in Table 2. The ROC curve (Figure 2) summarizes the comparison between the different modes. In group A, the NBI mode showed excellent sensitivity (95.0%), which was higher than the WLE 2 mode sensitivity (77.5%) and equal to WLE 1 + 2 mode sensitivity (95.0%). However, the greatest differences were recorded regarding specificity; NBI and WLE 1 + 2 + NBI showed a 96.8% and 94.2% specificity respectively, while in WLE 1 + 2 the specificity dropped to 40.6%. We observed also clear differences regarding the PPV (88.3% vs. 29.2%) and accuracy (96.4% vs. 51.8%) comparing NBI and WLE 1 + 2 modes. Finally, AUC summarizes the results described above, with excellent values for NBI (0.96, 95% CI 0.91–0.99) and WLE 1 + 2 + NBI (0.92, 95% CI 0.86–0.97), and significantly poorer accuracy in the WLE 1 + 2 mode (0.67, 95% CI 0.59–0.75).

No biopsy was performed in 125 out of 195 (64.1%) lesions and patients were followed up endoscopically; moreover, seven lesions were present after biopsy, for a total of 132 lesions addressed to follow-up. After a mean follow-up of 12.21 ± 4.63 months, two out of 132 (1.5%) lesions progressed to carcinoma and one out of 132 (0.7%) to high-grade dysplasia.

### 3.2. Group B

Surgical biopsies were indicated and performed in 29 out of 51 (56.9%) lesions of group B. The histological examination of group B gave the following results:
benign (0), *n* = 8 (27.6%)low-grade dysplasia (1), *n* = 4 (13.8%) (Figure 3)high-grade dysplasia (2), *n* = 6 (20.7%)carcinoma in situ or invasive (3), *n* = 11 (37.9%)

In addition, we included two stable and one regressed lesion at follow-up as “negative” biopsies (mean follow-up of 21.5 ± 3.5 months). Table 3 resumes the values of sensitivity, specificity, PPV, NPV, accuracy and likelihood. The ROC curve summarizes the comparison between the different modes (Figure 4). In group B, the NBI mode showed a higher sensitivity (82.4%) compared to the WLE 2 mode (64.7%), but slightly lower compared to the WLE 1 + 2 mode (88.3%). The greatest differences were recorded about specificity: NBI and WLE 1 + 2 + NBI showed a value of 94.1%, while in WLE 1 + 2 the specificity dropped to 29.4%. We observed also clear differences regarding the PPV (87.5% vs. 38.4%) and accuracy (90.2% vs. 49.0%) comparing NBI with WLE 1 + 2 modes. Finally, AUC summarizes the results described above with good values obtained for NBI (0.88, 95% CI 0.78–0.98) and WLE 1 + 2 + NBI (0.85, 95% CI 0.74–0.96). WLE 1 + 2 + NBI mode showed excellent values, while the data of the WLE 1 + 2 mode (0.58, 95% CI 0.47–0.69) were significantly lower, with the lower limit of the 95% CI below the value of 0.5. No biopsy was performed in 22 out of 51 (43.1%) lesions and patients were followed up endoscopically; moreover, seven lesions were still present after biopsy, for a total of 29 lesions addressed to follow-up. After a mean follow-up of 14.87 ± 4.11 months, 6 out of 29 (20.7%) lesions progressed to carcinoma and 1 out of 29 (3.4%) to high-grade dysplasia.

## 4. Discussion

The results of the present study showed improved diagnostic accuracy of NBI during videolaryngoscopy compared with WLE alone in both groups (A and B). In fact, the probability of a positive NBI lesion compared with a negative NBI lesion, was 29.68 (group A) and 13.96 (group B) times higher to be histologically positive.

The results obtained in group A are in accordance with those reported in literature with a similar cohort of patients [1,9,10]. In particular, Vilaseca et al. [1] showed that accuracy improved from 74.1% with WLE to 88.9% with NBI and demonstrated additionally, a better PPV (NBI: 89% vs. WLE: 74%). In addition, Popek et al. [9] showed a PPV of 97.7% for NBI vs. 79.6% for WLE.

More interesting are the results of group B compared with data present in literature. Piazza et al., performed two studies [12,13] about the endoscopic evaluation of patients in post-radiotherapy follow-up with NBI, enrolling respectively 59 and 183 patients. Zabrodsky et al. [14] evaluated with NBI endoscopy 66 patients previously treated with radiotherapy for head and neck cancer, including 85% with laryngeal neoplasm―a patient cohort easily comparable with group B of our study. Even though, at first sight, our results demonstrate lower sensitivity and specificity values for NBI compared with the above-mentioned studies, it is necessary to consider factors that could have influenced the results of the studies. For example, among the limitations stated in Piazza et al.’s [13] studies was that they considered lesions without performing biopsy as “true negatives” despite the short duration of follow-up; in the second study [12] each patient, at least twice endoscopically evaluated, was considered true negative; however, this assumption could be insufficient to obtain reliable values of specificity and NPV. Similarly to our study, Zabrodsky et al. [14] considered persistently negative endoscopy, with an average follow-up duration of 10 months (range 5–24 months) as true negative. In our study, on the other side, we considered the lesions of group B without biopsy after a mean follow-up of 21.5 ± 3.5 months as negative biopsies. In this regard, it should be emphasized that to date in literature no reliable indications regarding adequate follow-up duration are recommended.

Moreover, the differences between the results of group A and group B appear evident. The reasons for this discrepancy should be ascribed to the difficulty of interpreting endoscopic data in patients who underwent previous radiotherapy, as already reported in the above-mentioned articles. In our opinion, the great inhomogeneity between treated (with history of radiotherapy) and untreated patients (without history of radiotherapy), may reduce the clinical relevance of sensitivity, specificity, PPV, NPV and diagnostic accuracy data of NBI technology, if not separately studied.

Finally, data analysis, concerning progression rate to carcinoma, showed a difference between the two groups, even though not statistically significant. This trend may represent a future field of research to evaluate the appropriateness of follow-up attitudes in the two groups.

The strength of our study are the large cohort and the division in two groups considering previous radiotherapy treatment. Several studies reported similar results, for nasopharynx, hypopharynx or larynx [15,16,17]. Our results present “real-life” data assessing NBI technology. It represents a useful tool for daily clinical practice, in which we analyze mucosal lesions on different sites of the head and neck, considering that NBI is a standardized technique for all of them [6,15,18].

However, some limitations of our study should be mentioned. The duration of the study (average: 29 months) represented a limitation as only for 10 patients (seven in group A and three in group B) persistent negativity, as a negative biopsy, could be considered; therefore, to confirm data, the follow-up period should be extended. In fact, a study by Zhang et al. [19] suggests that patients with high-grade dysplasia should be followed up during the first three years, a period in which most malignant lesions generally progress. Another potential limitation, when evaluating the performance of the NBI, is that the filter is never used alone, but always after white light mode or alternation. A study protocol comparing the diagnostic capacity of NBI and WLE individually could precisely define the diagnostic accuracy of each method. Nevertheless, the use of NBI in association with WLE, as reported in our study and in literature, guarantees a more realistic assessment of its performance in daily clinical practice. Another feature that should be considered is the potential role of the learning curve, which was found to be the leading cause of false positives in the work conducted by Tirelli et al. [20]. Furthermore, this technology could have a positive impact on health care costs. In fact, detailed NBI endoscopic examination could reduce the number of unnecessary biopsies and identify early-stage lesions suitable for minimally invasive surgery, which could decrease hospital admissions.

## 5. Conclusions

Early diagnosis of laryngeal neoplasms remains a priority in otolaryngology with significant therapeutic and prognostic consequences. The technological evolution allows gaining additional information about biological characteristics to diagnose lesions, which cannot be obtained with traditional methods. Our high NPV demonstrates that NBI technology, which is capable of a detailed topographical definition of the lesion enhancing visualization of microvascular anomalies, represents a reliable instrument in challenging situations, especially in the context of post-radiotherapy or post-surgical mucosal changes.

## Figures and Tables

**Figure 1 jcm-10-01224-f001:**
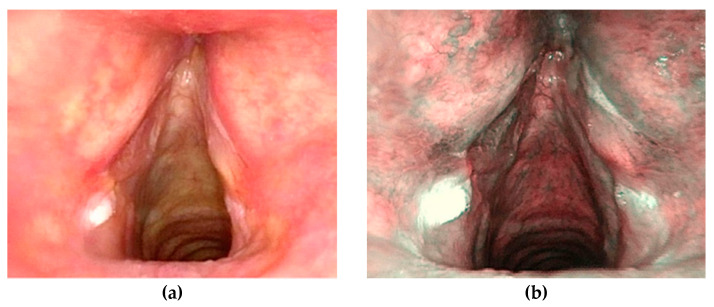
Patient, 63 years old with hyperkeratosic lesions, stable at three year close follow up. WLE endoscopy (**a**), NBI examination (**b**)**.**

**Figure 2 jcm-10-01224-f002:**
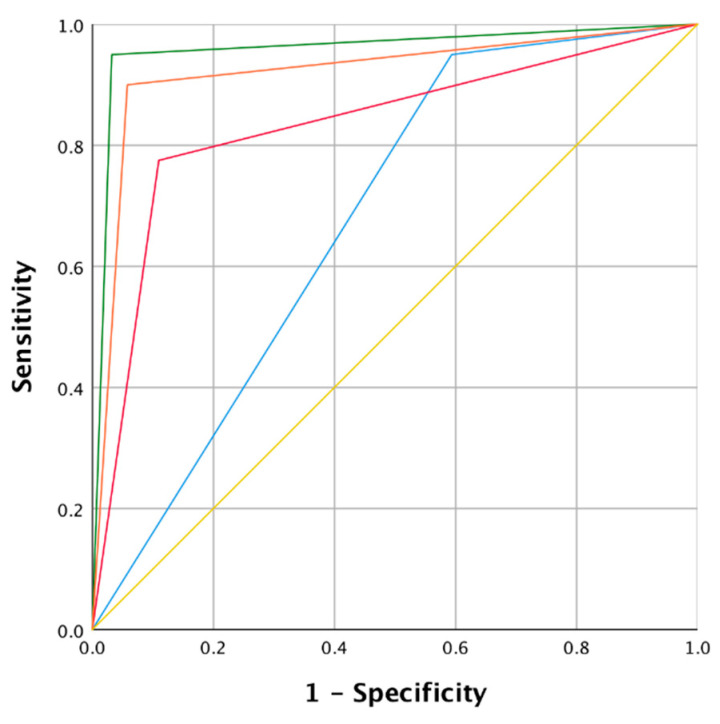
Receiver operating curve (ROC) and area under the curve (AUC) of the four modalities analyzed for group A. Colors of the curves: Green: narrow-band imaging (NBI); Orange: white light endoscopy (WLE) 1 + 2 + NBI; Red: WLE 2; Blue: WLE 1 + 2; Yellow: mean values.

**Figure 3 jcm-10-01224-f003:**
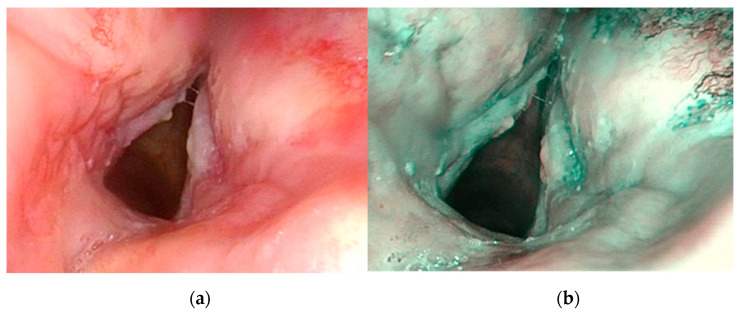
Patient, 65 years old, with history of laryngeal squamous cell carcinoma treated with chemoradiotherapy. Endoscopic appearance at one year follow-up. White light endoscopy (WLE) (**a**), Narrow band imaging (NBI) examination (**b**). Post-attinic microvascular alterations were suspicious for dysplasia and confirmed at histopathological exam.

**Figure 4 jcm-10-01224-f004:**
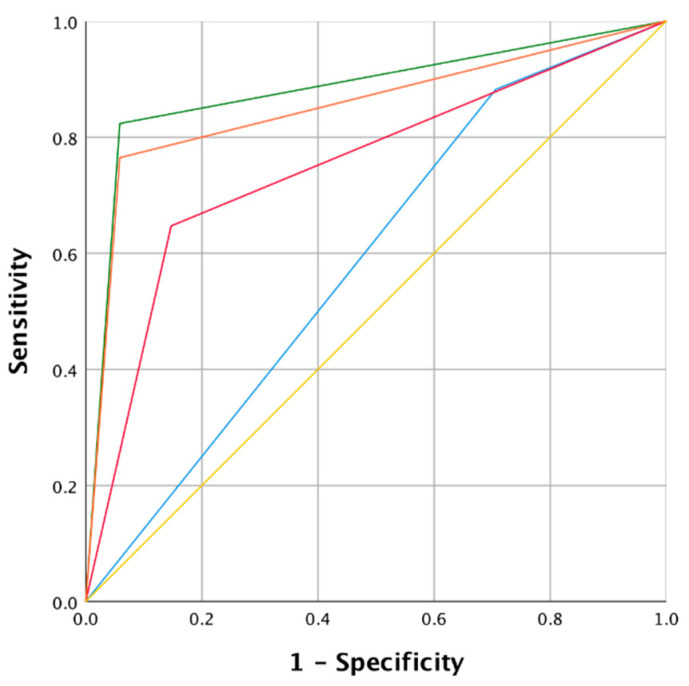
Receiver operating curve (ROC) and area under the curve (AUC) of the four modalities analyzed for group B. Colors of the curves: Green: NBI; Orange: WLE 1 + 2 + NBI; Red: WLE 2; Blue: WLE 1 + 2; Yellow: mean values.

**Table 1 jcm-10-01224-t001:** Patients’ demographic information

	Overall (*n* = 196)	Group A (*n* = 156)	Group B (*n* = 40)
Age ± SD (range)	60.5 ± 12.3 (26–93)	57.3 ± 13.4 (26–87)	62.8 ± 11.5 (32–93)
Sex (M–F)	145–51	112–39	33–12
Pts with >1 lesion	50	39	11
Total lesions	246	195	51

F: female; M: male; Pts: patients; SD: Standard Deviation.

**Table 2 jcm-10-01224-t002:** Sensitivity, specificity, positive predictive value (PPV), positive negative value (PNV), accuracy, likelihood ratio and comparison of area under the ROC curves (AUC) of the four diagnostic modalities in group A.

Indicator	NBI	WLE 1 + 2	WLE 2	WLE 1 + 2 and NBI
Sensitivity	95.0 %	95.0 %	77.5 %	90.0 %
Specificity	96.8 %	40.6 %	89.0 %	94.2 %
PPV	88.3 %	29.2 %	64.6 %	80.0 %
NPV	98.7 %	96.9 %	93.9 %	97.3 %
Accuracy	96.4 %	51.8 %	86.7 %	93.3 %
Positive likelihood ratio	29.68	1.59	7.04	15.51
Negative likelihood ratio	0.05	0.13	0.25	0.10
Mode	Area	Lower limit 95% CI	Upper limit 95% CI	*p*
NBI	0.96	0.91	0.99	<0.0001
WLE 1 + 2	0.67	0.59	0.75	<0.0001
WLE 2	0.83	0.75	0.91	<0.0001
WLE 1 + 2 and NBI	0.92	0.86	0.97	<0.0001
Mode comparison	z	Lower limit 95% CI	Upper limit 95% CI	*p*
WLE 1 + 2 vs. WLE 2	−4.168	−0.227	−0.082	<0.0001
WLE 1 + 2 vs. NBI	−8.699	−0.344	−0.217	<0.0001
WLE 1 + 2 vs. WLE + NBI	−6.608	−0.315	−0.171	<0.0001
WLE 2 vs. NBI	−3.420	−0.0199	−0.054	<0.001
WLE 2 vs. WLE + NBI	−2.091	−0.171	−0.006	0.037
NBI vs. WLE + NBI	2.040	0.001	0.074	0.041

CI: confidence interval; NBI: narrow band imaging; NPV: negative predictive value; PPV: positive predictive value; WLE: white light endoscopy.

**Table 3 jcm-10-01224-t003:** Sensitivity, specificity, positive predictive value (PPV), positive negative value (PNV), accuracy and likelihood ratio of the four diagnostic modalities applied in group B.

Indicator	NBI	WLE 1 + 2	WLE 2	WLE 1 + 2 and NBI
Sensitivity	82.4 %	88.3 %	64.7 %	76.5 %
Specificity	94.1 %	29.4 %	85.3 %	94.1 %
PPV	87.5 %	38.4 %	68.7 %	86.7 %
NPV	91.4 %	83.3 %	82.8 %	88.9 %
Accuracy	90.2 %	49.0 %	78.4 %	88.2 %
Positive likelihood ratio	13.96	1.25	4.40	12.97
Negative likelihood ratio	0.18	0.40	0.41	0.25
Mode	Area	Lower limit 95% CI	Upper limit 95% CI	*p*
NBI	0.88	0.78	0.98	<0.0001
WLE 1 + 2	0.58	0.47	0.69	<0.0001
WLE 2	0.75	0.62	0.88	<0.0001
WLE 1 + 2 and NBI	0.85	0.74	0.96	<0.0001
Mode comparison	z	Lower limit 95% CI	Upper limit 95% CI	*p*
WLE 1 + 2 vs. WLE 2	−2.365	0.347	−0.296	0.018
WLE 1 + 2 vs. NBI	−3.714	0.330	−0.449	<0.0001
WLE 1 + 2 vs. WLE + NBI	−3.674	0.335	−0.406	<0.0001
WLE 2 vs. NBI	−1.842	0.342	−0.273	0.066
WLE 2 vs. WLE + NBI	−1.532	0.348	−0.235	0.126
NBI vs. WLE + NBI	1.000	0.322	−0.028	0.317

CI: confidence interval; NBI: narrow band imaging; NPV: negative predictive value; PPV: positive predictive value; WLE: white light endoscopy.

## Data Availability

The data presented in this study are available on request from the corresponding author.

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
