# Peer review of "Role of Narrow Band Imaging Technology in the Diagnosis and Follow up of Laryngeal Lesions: Assessment of Diagnostic Accuracy and Reliability in a Large Patient Cohort"

_jcm, 2021, doi:10.3390/jcm10061224_

Round 1
Reviewer 1 Report
The abstract:
The abstract presents information in a somewhat chaotic way. Try to improve "flow" to make it clearer. The titles of individual sections would also be useful (Background, Material and methods, Results, Conclusions).
Abstract lacks information did reported differences were statistically significant? (provide at least p vales).
In the introduction:
The first reference is made after quite a large part of the text which provide a ;lot of information - I suggest adding some references.
Introduce full names for all abreviations at first mention also in the abstract (e.g. NBI). Then use these abbreviations consistently (e.g. UADT in the introduction is introduced twice; NBI in full in Materials and Methods, and more...).
Please use abreviations also for other... e.g. head and neck cancer etc.
In Materials and methods:
in sentence '... and patients that had previous treatment for..."
"Treatment" should be clearly defined.
Results:
Table 1 is not informative enough. Should be enriched in basic demographic and clinical information. Also comparison regard distribution of this variables between group A and B should be performed (p values should be provided).
Statistical difference between results of diagnostic utility of compared methods (for each group: A and B) should be be added to table 2 and 3 and to the text of Results section.
Figures 2 and 3 should include also p values.
Discussion:
In discussion please report the most important results (e.g. sensitivity, specyficyty, AUC and 95%CI, accuracy, PPV, NPV etc.) of diagnostic utility obtained by other Autors to facilitate results comparision.
In Conclusions:
This part should be more concise and should directly reflect aim of the study. I propose to move the remaining Authors' toughts, not drawed directly based on the results obtained, in the discussion.
Author Response
RESPONSE TO REVIEWER 1
Point 1:
The abstract:
The abstract presents information in a somewhat chaotic way. Try to improve "flow" to make it clearer. The titles of individual sections would also be useful (Background, Material and methods, Results, Conclusions).
Abstract lacks information did reported differences were statistically significant? (provide at least p vales).
Response: We modified the abstract as required, we presented information using the title for the sections and we added information about p-values in Results.
Point 2:
In the introduction:
The first reference is made after quite a large part of the text which provide a ;lot of information - I suggest adding some references.
Response: We modified the references as suggested by the reviewer.
Point 3:
Introduce full names for all abreviations at first mention also in the abstract (e.g. NBI). Then use these abbreviations consistently (e.g. UADT in the introduction is introduced twice; NBI in full in Materials and Methods, and more...).
Please use abreviations also for other... e.g. head and neck cancer etc.
Response: We modified the abbreviations in the text as required.
Point 3:
In Materials and methods:
in sentence '... and patients that had previous treatment for..."
"Treatment" should be clearly defined.
Response: We modified the sentence and we defined “treatment” as “previous radiochemotherapy treatment”.
Point 4:
Results:
Table 1 is not informative enough. Should be enriched in basic demographic and clinical information. Also comparison regard distribution of this variables between group A and B should be performed (p values should be provided).
Statistical difference between results of diagnostic utility of compared methods (for each group: A and B) should be be added to table 2 and 3 and to the text of Results section.
Figures 2 and 3 should include also p values.
Response: In Results, we modified tables and figures, and we added the p-values also for Mode Comparison and ROC curve.
Point 5:
Discussion:
In discussion please report the most important results (e.g. sensitivity, specyficyty, AUC and 95%CI, accuracy, PPV, NPV etc.) of diagnostic utility obtained by other Autors to facilitate results comparision.
Response: In Discussion, we reported the most important results of diagnostic utility and we have highlighted the clear differences regarding sensitivity, the PPV and accuracy, comparing NBI with WLE modes for both groups.
Point 6:
In Conclusions:
This part should be more concise and should directly reflect aim of the study. I propose to move the remaining Authors' toughts, not drawed directly based on the results obtained, in the discussion.
Response: We modified the conclusions as suggested by the reviewer.
Reviewer 2 Report
The authors evaluate in important subject: differences in sensitivity, specificity, PPV , NPV , accuracy between treated and untreated lesions of the UADT when using WLE and NBI.
However I have some questions and remarks.
- English is poor and text is not edited properly, unfortunately hampering the interpretation of the manuscript (examples: sensibility (tables), predictive negative value (line 70), not ondergone to biopsy (line 136), different colours of the lines in both figures without explanation, district (do you mean area?, one 5 lines (!) sentence at the end of the introduction)
- abbreviations are explained twice (like UADT line 33 and 58)
- The authors state they included 231 patients (after NBI endoscopy) with a macroscopic lesion of the mucosa of UADT, with endoscopic signs of cancer transformation or precancerous characteristics. Why was a biopsy with histopathological examination not performed in 229 - 91 = 138 lesions suspicious for malignancy after NBI endoscopy (inclusion criterion)? The same question regarding the 63 - 36 = 27 lesions from group B?
- the authors did not include polyps, but were there no patients with signs of papilloma or hyperkeratosis (sometimes difficult to distinguish from carcinoma (in situ)?
- I suggest to remove all non-larynx lesions from the analysis, since NBI interpretation of these sites is not comparable and introduces bias in the analysis. Nasopharynx, oropharynx and hypopharynx lesions are only a small group (34/195 group A and 12/63 group B) compared to larynx.
After adjusting the paper according to the given suggestions, I will continue reading the Discussion section.
Author Response
RESPONSE TO REVIEWER 2
Point 1
- English is poor and text is not edited properly, unfortunately hampering the interpretation of the manuscript (examples: sensibility (tables), predictive negative value (line 70), not ondergone to biopsy (line 136), different colours of the lines in both figures without explanation, district (do you mean area?, one 5 lines (!) sentence at the end of the introduction)
Response: The manuscript has been carefully revised by a professional language service, we modified the text as required, and we improved figures legend.
Point 2
- abbreviations are explained twice (like UADT line 33 and 58)
Response: We modified the abbreviations in the text.
Point 3
- The authors state they included 231 patients (after NBI endoscopy) with a macroscopic lesion of the mucosa of UADT, with endoscopic signs of cancer transformation or precancerous characteristics. Why was a biopsy with histopathological examination not performed in 229 - 91 = 138 lesions suspicious for malignancy after NBI endoscopy (inclusion criterion)? The same question regarding the 63 - 36 = 27 lesions from group B?
Response: In the new version, we modified patient population. In some cases, biopsies were not performed as there was not a clear clinical indication or if general status of patients did not allow it.
Point 4
- the authors did not include polyps, but were there no patients with signs of papilloma or hyperkeratosis (sometimes difficult to distinguish from carcinoma (in situ)?
Response: We decided to not include patients that received the clinical diagnosis of benign lesions, such as laryngeal polyps, cysts, Reinke's edema or vocal nodules, in case of absence of suspicious signs (i.e. hyperkeratosis, papillomatosis) during endoscopic examination.
Point 5
- I suggest to remove all non-larynx lesions from the analysis, since NBI interpretation of these sites is not comparable and introduces bias in the analysis. Nasopharynx, oropharynx and hypopharynx lesions are only a small group (34/195 group A and 12/63 group B) compared to larynx.
Response: In the new version of the manuscript, we removed all patients with non-laryngeal lesions from the study population and we presented the new results, as suggested by the reviewer.